# Prevalence of *Mycobacterium leprae* in armadillos in Brazil: A systematic review and meta-analysis

**Patrícia Deps[1,2]\*, João Marcelo Antunes[3], Adalberto Rezende Santos[4], Simon M. Collin**[5]

**1** Department of Social Medicine, Universidade Federal do Espírito Santo, Vitória, Espírito Santo, Brazil,
**2** Postgraduate Programme in Infectious Diseases, Universidade Federal do Espírito Santo, Vitória, Espírito Santo, Brazil, **3** Universidade Federal Rural do Semi-Árido, Hospital Veterinário Jerônimo Dix-Huit Rosado Maia, Mossoró, Rio Grande do Norte, Brazil, **4** Laboratório de Biologia Molecular Aplicada a Micobactérias, Instituto Oswaldo Cruz (IOC/Fiocruz), Rio de Janeiro, Brazil, **5** National Infection Service, Public Health England, London, United Kingdom

\* patricia.deps@ufes.br

**Data Availability Statement:** All relevant data are within the manuscript and its Supporting Information files.

**Funding:** This study received no funding.

## Abstract

Understanding the prevalence of *M. leprae* infection in armadillos is important because of evidence from Brazil and other countries of an association between contact with armadillos and the development of Hansen's Disease (leprosy). Our aim was to characterize studies which have investigated natural *M. leprae* infection in wild armadillos in Brazil, and to quantify and explore variability in the reported prevalence of infection. We conducted a systematic review (PROSPERO CRD42019155277) of publications in MEDLINE, EMBASE, Global Health, Scopus, LILACS, Biblioteca Digital Brasileira de Teses e Dissertações, Catálogo de Teses e Dissertações de CAPES, and Biblioteca Virtual em Saúde up to 10/2019 using Mesh and text search terms (in English, Portuguese, Spanish, and French). The 10 included studies represented a total sample of 302 armadillos comprising 207 (69%) *Dasypus novemcinctus*, 67 (22%) *Euphractus sexcinctus*, 16 (5%) *Priodontes maximus*, 10 (3%) *Cabassous unicinctus*, and 2 (1%) *Cabassous tatouay* from 7 different states. Methods used included histopathology (4 studies), PGL-1 and LID-1 antigen detection (4 studies) and examination for clinical signs of disease (4 studies). Eight studies used PCR of which 7 targeted the RLEP repetitive element and 3 tested for inhibitory substances. *M. leprae* prevalence by PCR ranged from 0% (in 3 studies) to 100% in one study, with a summary estimate of 9.4% (95% CI 0.4% to 73.1%) and a predictive interval of 0–100%. The average prevalence is equivalent to 1 in 10 armadillos in Brazil being infected with *M. leprae*, but wide variation in sample estimates means that the prevalence in any similar study would be entirely unpredictable. We propose instead that future studies aim to investigate transmission and persistence of *M. leprae* within and between armadillo populations, meanwhile adopting the precautionary principle to protect human health and an endangered species in Brazil.

**Competing interests:** The authors have declared that no competing interests exist.

## Author summary

The risk to human health of contact with armadillos infected with *Mycobacterium leprae*, a bacterium that causes Hansen's Disease (leprosy), is uncertain, but evidence from Brazil and other countries appears to show a link between contact with armadillos and increased risk of Hansen's Disease in people. How much of Hansen's Disease in the human population is caused by contact with armadillos will depend on the size of the risk, the type and frequency of contact and how common it is in the population, and the role of other (human-to-human) transmission routes for *Mycobacterium leprae*. Our review has shown that one other key factor, the proportion of wild armadillos infected with *Mycobacterium leprae*, cannot be predicted with any certainty based on data from studies conducted to date. We suggest that much bigger and longer-term studies are needed, perhaps in partnership with animal conservation and ecology groups, to map *Mycobacterium leprae* infection in armadillos across Brazil and correlate this with proximity to human habitats. At the same time, data must be gathered in studies focused on populations of armadillos to characterize *Mycobacterium leprae* transmission and persistence within groups of animals, for example, using trackers and repeated sampling over the animals' lifespans. In the meantime, the precautionary principle should prevail, and public health and educational efforts should be directed to improving community knowledge and changing behaviour to protect people and armadillos.

## Introduction

Understanding the prevalence of *M. leprae* infection in armadillos is of public health importance because of epidemiological evidence from Brazil and other countries of an association between contact with armadillos and the development of Hansen's Disease (leprosy) in people [1–5]. The first report of natural infection of *M. leprae* in wild armadillos in Brazil was a preliminary finding in 2002 based on PCR analysis of blood samples from 14 nine-banded armadillos (*Dasypus novemcinctus*) from the south-eastern state of Espírito Santo [6]. A later study confirmed these findings in Espírito Santo [7], and *M. leprae* was subsequently reported in wild armadillos from the northern states of Ceará [8] and Pará [3]. Conversely, studies in São Paulo and Mato Grosso do Sul [9] and Amazonas [10] found no *M. leprae* in wild armadillos.

Brazil is a high-burden country for Hansen's Disease [11], with incidence varying according to geographic and socioeconomic determinants [12]. Although the disabling and disfiguring sequelae of Hansen's Disease are entirely avoidable if diagnosed and treated early [13], the social stigma of 'leprosy' has not been entirely dispelled, and still has a profoundly negative impact on people diagnosed with this disease [14, 15]. The proportion of new cases in Brazil attributable to zoonotic transmission is unknown, but armadillo capture and consumption persists as a cultural habit in some parts of Brazil and wild armadillos in these areas may serve as a reservoir of *M. leprae* infection in humans [16].

The aim of this systematic review and meta-analysis was to characterize studies which have investigated natural *M. leprae* infection in wild armadillos in Brazil, and to quantify and explore variability in the prevalence of infection.

## Methods

### Review protocol

The protocol for this systematic review was defined in advance and registered with PROSPERO (CRD42019155277). A PRISMA checklist is provided as S1 PRISMA checklist.

## Searches

We searched the following databases and libraries between October 26th-27th 2019: MEDLINE (Epub Ahead of Print, In-Process & Other Non-Indexed Citations and Daily 1946 to October 25, 2019), EMBASE (1974 to 2019 October 25), Global Health, Scopus, LILACS (Latin American and Caribbean Center on Health Sciences Information), Biblioteca Digital Brasileira de Teses e Dissertações (BDTD), Catálogo de Teses e Dissertações de CAPES (Coordenação de Aperfeiçoamento de Pessoal de Nível Superior), Biblioteca Virtual em Saúde (BVS). Full search terms are provided in the supplementary appendix. In brief, we used Mesh and text search terms for: ("Mycobacterium leprae" OR "Leprosy") AND ("Armadillos" OR "Dasypus novemcinctus" OR "Dasypus septecinctus" OR "Euphractus sexcinctus") AND "Brazil" in MEDLINE and EMBASE supplemented by Portuguese, Spanish and French equivalents in other databases (leprosy = lepra OR Hanseníase OR lepre; armadillo = tatu OR tatou; Brazil = Brasil OR Brésil). We imposed no date, language or publication type restrictions. Citations identified by the search were imported into EndNote (EndNote X9; Clarivate Analytics, Boston, MA 02210, USA) for de-duplication. Bibliographies of all included studies were searched manually.

## Screening, inclusion/exclusion, quality assessment and data extraction

Screening and quality assessment were conducted independently and in parallel by three reviewers: title and abstract SC and PD; full text SC and JMA; quality assessment SC, JMA and PD. References were included if they described *Mycobacterium leprae* carriage or natural infection in wild armadillos in Brazil, regardless of armadillo species or microbiological method. Studies involving experimental infection and/or involving animals that were already captive were excluded. The methodological quality of each included study was rated using a 10-item quality assessment tool adapted from the NIH Quality Assessment Tool for Observational Cohort and Cross-Sectional Studies (**S1 Data**) [17]. The adaptation allowed for the assessment of data quality pertaining specifically to animal pathogen carriage/infection studies. Each study was rated as being of 'good', 'fair' or 'poor' quality based on the average score of the two reviewers. Data extraction was done independently and in parallel by two reviewers (SC and JMA) into a spreadsheet (**S1 Data**). The primary outcome for data extraction was the proportion of the captured armadillos which tested positive for *M. leprae*. Other extracted variables included: geographical region of Brazil; armadillo species; diagnostic method (e.g. PCR, ELISA); specimen type (e.g. tissue, blood); and tissue type (e.g. liver, brain, skin, etc.).

## Analysis

Key characteristics of each included study were summarized qualitatively. Binomial-normal random effects meta-analysis of the proportion of captured armadillos in which *M. leprae* was detected was performed in Stata (StataCorp. 2017. Stata Statistical Software: Release 15. College Station, TX, USA) using *metaprop_one* [18]. In this approach, the binomial distribution is used to model within-study variability, and the normal distribution is used to model the random effects. Between-study heterogeneity was estimated as $\tau^2$, and evidence of heterogeneity was tested by Likelihood Ratio (LR) test comparing random and fixed effects models. The proportion of overall heterogeneity attributable to between-study variance was quantified using a formulation of the $I^2$ statistic for binary variables [19]. Prediction intervals were estimated to show the expected prevalence of *M. leprae* (% positive armadillos) accounting for between-study variability [20, 21]. Meta-analysis defaulted to fixed effects if 3 or fewer studies were included. We used Egger's test to detect small-study bias. Subgroup analyses specified *a priori* (subject to sufficient data) were by geographic region, armadillo species, diagnostic method, specimen type, and tissue type.

## Results

Database searches identified 122 references (**S1 PRISMA Flow Diagram**). A study known to be under review at the time of database searching was also included [22]. After de-duplication and screening by title and abstract, 13 references were retained for full text review, of which 10 were included for data extraction. Quality assessment rated 8 as 'good' and 2 as 'fair' quality (reviewer agreement 9/10) (**S1 Data**). Five of the full text articles assessed for eligibility were theses or dissertations [7, 23–25, 28]. The full PCR results from two of these had been published in peer-reviewed papers that we included for data extraction: Pedrini 2006 thesis [24] in Pedrini *et al* 2010 [9]; Portela 2015 dissertation [25] in da Silva *et al* 2018 [3]. PCR results from the Deps 2003 thesis [23] were published as preliminary findings in Deps *et al* 2002 [6]; PCR results from the Antunes 2007 dissertation [7] and de Souza dissertation [28] had not yet been published. Key features and findings of the 10 included studies are summarized in **Table 1**, with further details of each study provided in **S1 Table**.

**Table 1. Main characteristics and findings of included studies investigating *M. leprae* infection in wild armadillos in Brazil.**

| Author | Year | Location | Ref | Armadillo species | Total number caught | Test method(s)[a, b] | Positive animals (tissue) | Positive animals (blood) | Test method details[a] |
|---|---|---|---|---|---|---|---|---|---|
| Deps | 2003 | Espírito Santo | [23] | *Dasypus novemcinctus* | 52 | PCR[c] | 19/36 (53%) | 5/42 (12%) | single-round 65 kDa (+ Southern Blot Hybridization) |
| | | | | *Dasypus novemcinctus* | 52 | BCG, HE, ZN | 0/47 (BCG) 0/48 (HE) 0/50 (ZN) | | Ear tissue |
| Deps *et al* | 2007 | Espírito Santo | [26] | *Dasypus novemcinctus* | 52 | ILF | - | 11/37 (30%) | PGL-1 rapid test |
| Deps *et al* | 2008 | Espírito Santo | [27] | *Dasypus novemcinctus* | 66 | ELISA | - | 5/47 (11%) | PGL-1 IgM |
| Antunes | 2007 | Espírito Santo | [7] | *Dasypus novemcinctus* | 65 | PCR | 4/65 (6%) | - | single-round 18kDa, RLEP (+ qPCR + sequencing) |
| Pedrini *et al* | 2010 | São Paulo + Mato Grosso do Sul | [9] | *Dasypus novemcinctus* | 18 | PCR | 0/18 | 0/2 | single-round RLEP (+ MegaBACE 1000 sequencing) |
| | | | | *Euphractus sexcinctus* | 22 | PCR | 0/22 | 0/19 | single-round RLEP (+ MegaBACE 1000 sequencing) |
| | | | | *Cabassous tatouay* | 2 | PCR | 0/2 | 0/2 | single-round RLEP (+ MegaBACE 1000 sequencing) |
| | | | | *Cabassous unicinctus* | 2 | PCR | 0/2 | 0/2 | single-round RLEP (+ MegaBACE 1000 sequencing) |
| | | | | All species as above | 44 | ZN | 0/44 | - | |
| Frota *et al* | 2012 | Ceará | [8] | *Dasypus novemcinctus* | 27 | PCR | 5/27 (19%) | - | nested RLEP[inh] (+ *gyrA* sequencing[d]) |
| | | | | *Euphractus sexcinctus* | 2 | PCR | 1/2 (50%) | - | nested RLEP[inh] (+ *gyrA* sequencing[d]) |
| de Souza | 2016 | Mato Grosso do Sul | [28] | *Priodontes maximus* | 16 | PCR | 0/16 | - | single-round RLEP (+ qPCR + mPCR + VNTR) |
| | | | | *Euphractus sexcinctus* | 17 + 6[e] | PCR | 0/23 | - | single-round RLEP (+ qPCR + mPCR + VNTR) |
| | | | | *Dasypus novemcinctus* | 2 + 1[e] | PCR | 0/3 | - | single-round RLEP (+ qPCR + mPCR + VNTR) |
| | | | | *Cabassous unicinctus* | 8 | PCR | 0/8 | - | single-round RLEP (+ qPCR + mPCR + VNTR) |

(*Continued*)

**Table 1.** (Continued)

| Author | Year | Location | Ref | Armadillo species | Total number caught | Test method(s)[a, b] | Positive animals (tissue) | Positive animals (blood) | Test method details[a] |
|---|---|---|---|---|---|---|---|---|---|
| da Silva *et al* | 2018 | Pará | [3] | *Dasypus novemcinctus* | 16 | PCR[f] | 10/16 (63%) | - | single-round RLEP (+ WGS) |
| Stefani *et al* | 2019 | Amazonas | [10] | *Dasypus novemcinctus* | 12 | PCR | 0/12 | - | single-round RLEP[inh] |
| | | | | *Dasypus novemcinctus* | 12 | HE, FF[g] | 0/12 | - | |
| da Silva Ferreira *et al* | 2020 | Rio Grande do Norte | [22] | *Euphractus sexcinctus* | 20 | PCR | 20/20 | - | nested RLEP[inh] (+ RFLP) |
| | | | | *Euphractus sexcinctus* | 20 | ELISA, ILF[h] | - | 20/20 | PGL-1 IgM, LID-1 IgG |

[a] BCG = Bacillus Calmette–Guérin (antigen immunohistochemistry); ELISA = enzyme-linked immunosorbent assay; FF = Fite Faraco; HE = haematoxylin and eosin stain; ILF = immunochromatographic lateral flow test; PCR = Polymerase Chain Reaction (mPCR = multiplex PCR; qPCR = Real Time PCR); PGL-1 = phenolic glycolipid 1; RFLP = restriction fragment length polymorphism analysis; RLEP = *M. leprae*-specific repetitive element; VNTR = variable number tandem repeat (genotyping); WGS = whole genome sequencing; ZN = Ziehl-Neelsen (bacilloscopy)

[b] Four studies investigated clinical signs of leprosy, two with positive findings (Deps [23] and Antunes [7]), one negative (de Souza [28]) and one inconclusive (Stefani et al [10])

[c] Deps reported ML Flow rapid immunochromatographic serology (PGL-1) results in Deps *et al* 2007 [26] and ELISA (PGL-1) results in Deps *et al* 2008 [27]

[d] the analysed samples belonged to the gyrAT (SNP type 3) population, which was also identified in wild armadillos in the USA [29] and in humans in Brazil [30]

[e] roadkill animals

[f] da Silva *et al* used SYBR Gold and auramine/rhodamine staining techniques (staining of mycobacteria *in situ*), detection of PGL-1 antigen using polyclonal rabbit antibody and acid-fast staining of bacilli using HE and FF techniques in spleen sections from PCR-positive wild armadillos, but total numbers of samples tested using these techniques and overall concordance with PCR results was not reported [3]

[g] Following complete dermato-neurological examination by a dermatologist, skin lesions suspect of leprosy were biopsied. Skin sections were further prepared for histopathological examination after HE and FF staining for bacilli identification. 48 skin sections on 96 slides were tested, all were negative, but one armadillo showed skin histopathology compatible with paucibacillary leprosy, another showed granulomas with epithelioid and Langerhans cells [10]

[h] ELISA IgM against PGL-1 and IgG against LID-1 antigens; NDO-LID rapid ILF test (Orange Life, Rio de Janeiro, Brazil); ML Flow ILF test (acquired from Dr. Samira Bührer-Sékula, Royal Tropical Institute, KIT Biomedical Research, Amsterdam, the Netherlands)

[inh] inhibitory substances tested for in negative DNA samples

## Study sites and capture of armadillos

The geographical locations of the 10 included studies are shown in **Fig 1**. Four studies were based in Espírito Santo state in the south east region of Brazil, 3 of which (Deps and Deps *et al.*) used all or part of a total sample of 66 armadillos (all *Dasypus novemcinctus*) collected between June 2000 and July 2001 [23, 26, 27], one (Antunes) a later sample of 69 *Dasypus novemcinctus* caught mainly in a different part of the state between July 2004 and July 2005 [7].

Two studies were based in the adjacent São Paulo (south east Region) and/or Mato Grosso do Sul (central west region) states, the 2010 study collecting a sample of 44 armadillos (mainly *Dasypus novemcinctus* and *Euphractus sexcinctus*) from both states at unspecified dates [9], the 2016 study a sample of 43 live (mostly *Euphractus sexcinctus* and *Priodontes maximus*) and 7 roadkill armadillos collected between June 2011 and January 2015 from the same ecoregion (Pantanal da Nhecolândia) in Mato Grosso do Sul as the earlier study [28].

The four remaining studies were located in the north or north east of Brazil: a 2012 study caught 29 armadillos (27 *Dasypus novemcinctus*, 2 *Euphractus sexcinctus*) between July and August 2007 in the north east region state of Ceará [8]; two more recent studies caught 16 and 12 *Dasypus novemcinctus* from the states of Pará (unspecified dates) and Amazonas (expedition in August 2015) [10], respectively; the most recent study (2019) caught 20 *Euphractus*

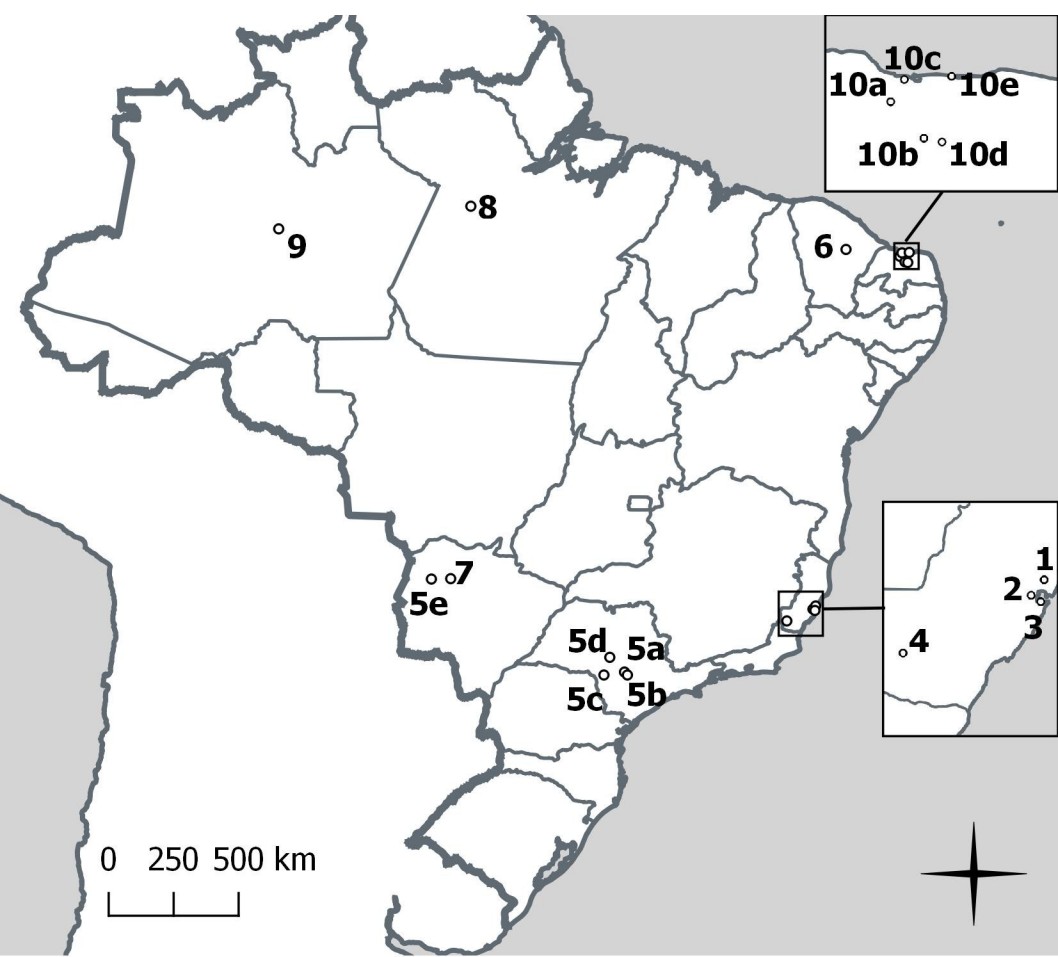

**Fig 1. Locations of studies investigating *M. leprae* infection in wild armadillos in Brazil.** 1—Deps, 2003, Espírito Santo; 2—Deps *et al.*, 2007, Espírito Santo; 3—Deps et al., 2008, Espírito Santo; 4—Antunes, 2007, Espírito Santo; 5—Pedrini et al., 2010, São Paulo and Mato Grosso do Sul; 6—Frota et al., 2012, Ceará; 7—de Souza, 2016, Mato Grosso do Sul; 8—da Silva et al., 2018, Pará; 9—Stefani et al., 2019, Amazonas; 10a-10e - da Silva Ferreira et al., 2020, Rio Grande do Norte. Map produced using QGIS, Open Source Geospatial Foundation Project http://qgis.osgeo.org.

*sexcinctus* between May and June 2016 in the north east region state of Rio Grande do Norte [22].

The 10 included studies yielded a total sample of 302 armadillos (295 live, 7 roadkill), comprising 207 (69%) *Dasypus novemcinctus* ('Nine-banded'), 67 (22%) *Euphractus sexcinctus* ('Six-banded'), 16 (5%) *Priodontes maximus* ('Giant'), 10 (3%) *Cabassous unicinctus* ('Southern Naked-tailed'), and 2 (1%) *Cabassous tatouay* ('Greater Naked-tailed') (**Fig 2**). Armadillos were captured by local hunters in 7 studies [3, 8, 10, 22, 23, 26, 27], by veterinarians or wildlife specialists in 2 studies [7, 28] and the method was not reported in one study [9].

## Biological samples

Armadillos were anaesthetized and euthanised in 5 studies [7–10, 22], captured and released in 4 studies [23, 26–28] or specimens were obtained from animals recently killed by local hunters [3]. Blood specimens were taken in 7 studies [7, 9, 22, 23, 26–28] but only three authors (Deps [23, 26, 27], Pedrini *et al* [9], da Silva Ferreira *et al* [22]) used these to test for *M. leprae*. A wide range of tissue types were sampled (**S1 Data**), with all except 2 studies (Pedrini *et al*

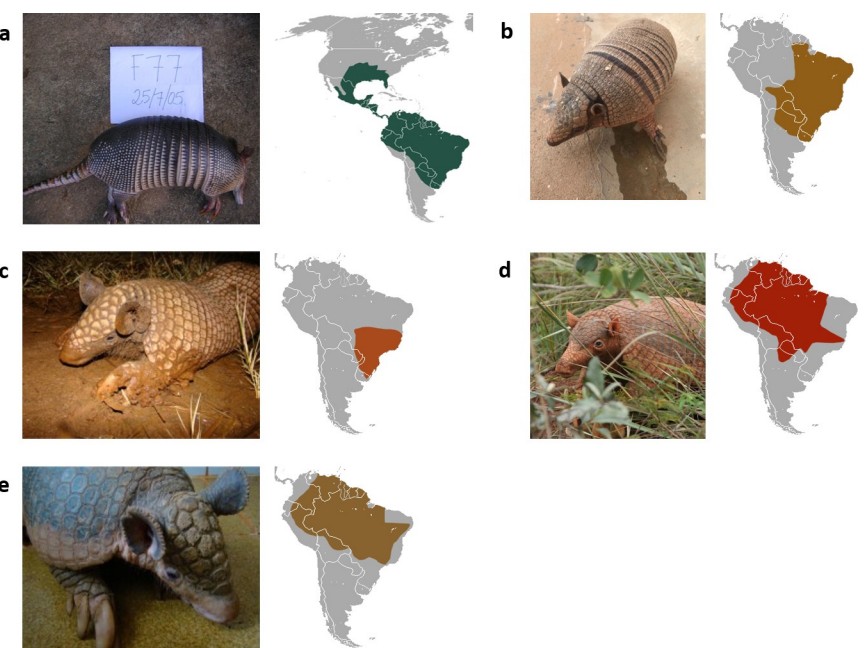

**Fig 2. Armadillo species investigated for natural *M. leprae* infection in Brazil with geographic distributions.** a) *Dasypus novemcinctus* (Nine-banded armadillo, common long nosed armadillo); b) *Euphractus sexcinctus* (Six-banded armadillo, yellow armadillo); c) *Cabassous tatouay* (Greater naked-tailed armadillo); d) *Priodontes maximus* (Giant armadillo); e) *Cabassous unicinctus* (Southern naked-tailed armadillo, leathered tail armadillo). Credits: a) the authors (JMA); b) Laboratory of Studies in Immunology and Wildlife at UFERSA, Mossoró-RN, Brazil; c) & e) Instituto Chico Mendes de Conservação da Biodiversidade (ICMBio), Brazil; d) Carly Vynne; all species distribution maps from Wikimedia commons, attribution www.iucnredlist.org, accessed 11/12/2019.

[9], da Silva Ferreira et al [22]) using spleen specimens and all except 2 (Stefani *et al* [10], da Silva *et al* [3]) using liver and/or ear tissue specimens. One study (Pedrini *et al* [9]) also tested for *M. leprae* in one faeces specimen and a small number (5) of nostril swabs.

## DNA

Test results using PCR to detect *M. leprae* DNA were reported in all studies except Deps *et al* 2007 [26] and Deps *et al* 2008 [27] but this author had reported preliminary PCR results in 2002 [6] and full PCR results were included in our meta-analysis [23]. All but one of the 8 PCR studies targeted the RLEP *M. leprae*-specific repetitive element [3, 7–10, 28]; Deps targeted a 372bp groE-L gene sequence encoding the 65kDa protein [23] and Antunes (in addition to RLEP) targeted a 360bp sequence encoding the 18kDa protein [7]. The 7 RLEP studies differed with respect to PCR methods (primers, testing for inhibitory substances, nested PCR) and whether positive PCR results were confirmed as *M. leprae* by other genomic methods (sequencing, VNTR, RFLP) (**Table 1**).

Three of the 5 studies which used a single round of PCR for RLEP reported 0% positivity [9, 10, 28], with the other two reporting 6% (4/65) and 63% (10/16) positivity [3, 7]; the two studies which used nested PCR for RLEP reported 21% (6/29) and 100% (20/20) positivity [8, 22]. Inhibitory substances were tested for in 3 studies [8, 10, 22], being detected in 0/12, 1/29, 1/20 of samples where the corresponding PCR results were 0% (0/12), 21% (6/29) and 100% (20/20), respectively. The two non-RLEP PCR studies reported 53% (19/36) and 3% (2/65) positivity based on 65kDa conventional and 18kDa real-time PCR, respectively [7, 23]. The two 18kDA-positive samples were among 4 samples which were RLEP-positive [7].

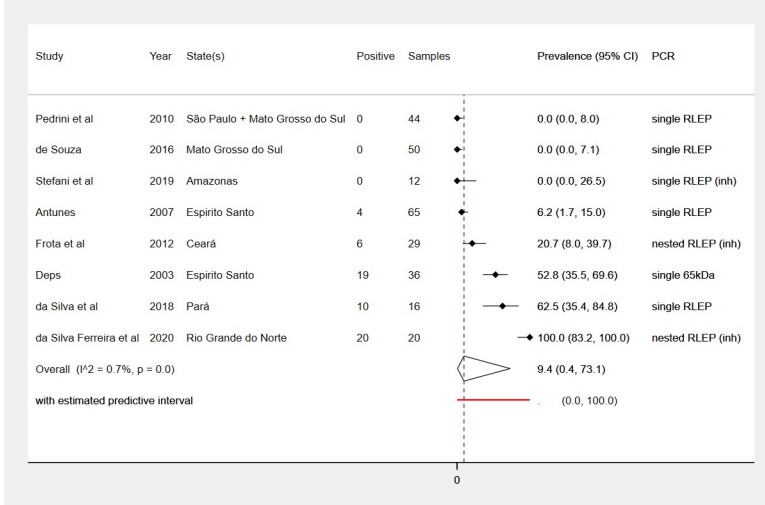

| Study | Year | State(s) | Positive | Samples | Prevalence (95% CI) | PCR |
|-------|------|----------|----------|---------|--------------------|-----|
| Pedrini et al | 2010 | São Paulo + Mato Grosso do Sul | 0 | 44 | 0.0 (0.0, 8.0) | single RLEP |
| de Souza | 2016 | Mato Grosso do Sul | 0 | 50 | 0.0 (0.0, 7.1) | single RLEP |
| Stefani et al | 2019 | Amazonas | 0 | 12 | 0.0 (0.0, 26.5) | single RLEP (inh) |
| Antunes | 2007 | Espírito Santo | 4 | 65 | 6.2 (1.7, 15.0) | single RLEP |
| Frota et al | 2012 | Ceará | 6 | 29 | 20.7 (8.0, 39.7) | nested RLEP (inh) |
| Deps | 2003 | Espírito Santo | 19 | 36 | 52.8 (35.5, 69.6) | single 65kDa |
| da Silva et al | 2018 | Pará | 10 | 16 | 62.5 (35.4, 84.8) | single RLEP |
| da Silva Ferreira et al | 2020 | Rio Grande do Norte | 20 | 20 | 100.0 (83.2, 100.0) | nested RLEP (inh) |
| Overall  (I^2 = 0.7%, p = 0.0) | | | | | 9.4 (0.4, 73.1) | |
| with estimated predictive interval | | | | | (0.0, 100.0) | |

**Fig 3. Prevalence of natural *M. leprae* infection in wild armadillos in Brazil detected using PCR methods.**
PCR = Polymerase Chain Reaction; RLEP = M. leprae-specific repetitive element; (inh) = inhibitory substances tested for in negative DNA samples.

Other genomic methods to confirm PCR results were described in 6 studies but were not used in two of these studies because PCR results were negative [9, 28]. In the remaining 4 studies, PCR results were confirmed to be *M. leprae* in 2/4 positive samples by RLEP copy sequence [7], in 6/6 by *gyrA* gene sequence [8], in 10/10 by RLEP sequence [3], and in 20/20 by RFLP [22].

Overall test results (*M. leprae* PCR-positive) from the 8 PCR studies are summarized in **Table 1** and **Fig 3**. *M. leprae* prevalence by PCR ranged from 0% in 3 studies (12 *Dasypus novemcinctus* in Amazonas state [10], 50 armadillos of various species in Mato Grosso do Sul [28], 44 of various species in Mato Grosso do Sul and São Paulo [9]) to 100% in 20 *Euphractus sexcinctus* in Rio Grande do Norte state [22]. The other four PCR studies reported prevalences of 6% in 65 *Dasypus novemcinctus* in Espírito Santo [7], 21% in 29 mostly *Dasypus novemcinctus* in Ceará [8], 53% in 36 *Dasypus novemcinctus* in Espírito Santo [23], and 63% in 16 *Dasypus novemcinctus* in Pará state [3].

The summary estimate for *M. leprae* prevalence was 9.4% (95% CI 0.4% to 73.1%) (**Fig 3**), with between-study heterogeneity ($\tau^2$ = 17.7) representing a negligible proportion of overall variance in this estimate ($I^2$ = 1%, p<0.01). The predictive interval shows that the prevalence in a hypothetical future study with characteristics similar to the included studies would be entirely unpredictable, i.e. prevalence could be between 0% and 100%.

Egger's regression test for funnel plot asymmetry did not indicate small-study bias (p = 0.16) although the 100% prevalence study was outside the pseudo 95% confidence limits (**S1 Fig**).

Differences in prevalence by tissue type in studies which found non-zero prevalence and tested multiple tissues types is shown in **Fig 4**. Moderate heterogeneity between tissue types within the Deps study was weakly supported by statistical evidence ($I^2$ = 35%, p = 0.1); there was no heterogeneity between tissue types for Frota *et al* ($I^2$ = 0%, p = 1.0).

*M. leprae* prevalence (detected by PCR) in the two most commonly sampled armadillo species (*Dasypus novemcinctus* and *Euphractus sexcinctus*) is shown in **Fig 5**. All other sampled armadillo species had zero prevalence, *Cabassous unicinctus* (0/2 and 0/8 animals) [9, 28], *Cabassous tatouay* (0/2) [9] and *Priodontes maximus* (0/16) [28], but these results were

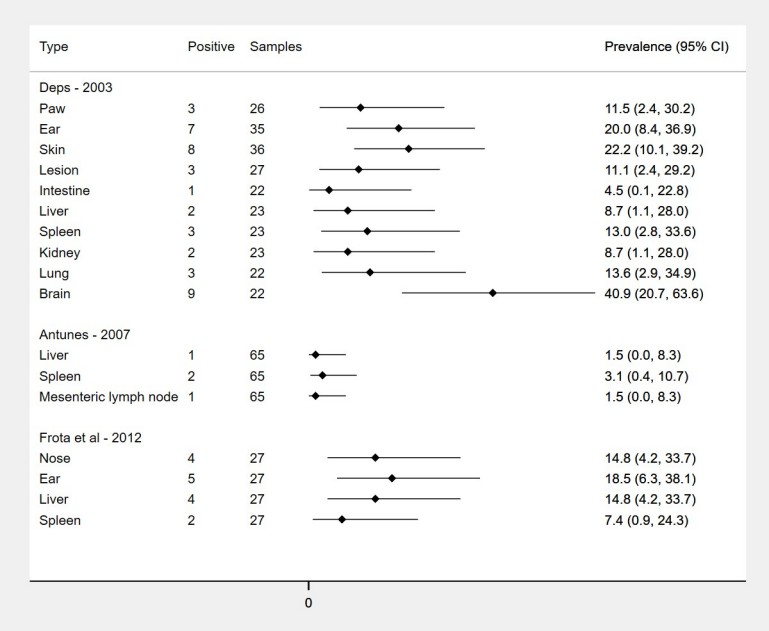

**Fig 4. Prevalence of natural *M. leprae* infection in wild armadillos in Brazil by tissue type.**

reported by studies which also found no *M. leprae* in *Dasypus novemcinctus* and *Euphractus sexcinctus*. The included studies did not provide sufficient data to support meaningful sub-group analyses by geographic region, diagnostic method or specimen type.

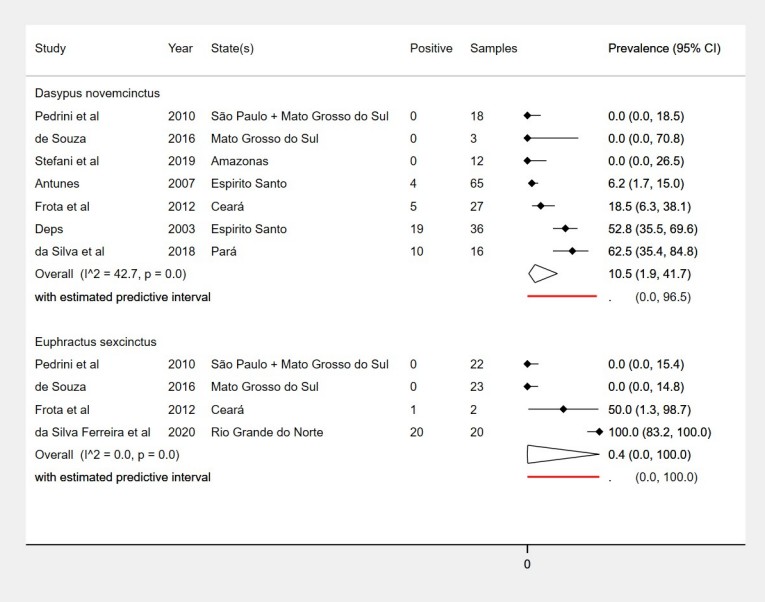

**Fig 5. Prevalence of natural *M. leprae* infection in wild armadillos in Brazil by species.**

## Histopathology

Four studies used staining techniques including Ziehl-Neelsen (ZN), haematoxylin and eosin (HE) and Fite Faraco (FF) to identify *M. leprae* in tissue samples: Deps 2003 (ZN, HE) [23]; Pedrini *et al* (ZN) [9]; da Silva *et al* (HE, FF, SYBR Gold, and auramine/rhodamine) [3]; and Stefani *et al* (HE, FF) [10]. Deps 2003 also used Bacillus Calmette-Guérin (BCG) antigen immunohistochemistry [23] and Pedrini *et al* inoculated liver, spleen and mesenteric lymph node specimens in LJ culture medium [9].

Deps reported 0/50 and 0/47 positive results in ear tissues samples by ZN and BCG techniques, respectively [23]. By contrast, HE results showed scarce infiltration in 15/48 (31%) ear tissue samples, moderate in 21/48 (44%) and intense in 10/48 (21%).

Pedrini *et al* reported entirely negative ZN results, consistent with their PCR results [9]. Stefani *et al* reported negative FF acid-fast bacilli results in 48 specimens from 12 armadillos, whilst HE stained tissue sections did not show histopathological features of *M. leprae* infection except for one skin fragment that presented unspecific inflammatory infiltrate suggestive of indeterminate leprosy [10]. Da Silva *et al* reported positive histopathological findings in tissue specimens from animals that had tested positive for *M. leprae* by PCR, but did not report overall positivity by each technique in all of the armadillos in their sample (10 PCR-positive, 6 PCR-negative) [3].

## PGL-1 and LID-1

Three studies tested for *M. leprae* phenolic glycolipid 1 (PGL-1) antigen [3, 26, 27], and one study tested for both PGL-1 and LID-1 reactivity [22]. Deps *et al* 2007 used ML Flow rapid immunochromatographic serology [26] and ELISA [27], reporting 11/37 (47%) positive by ML Flow compared with 5/47 (11%) positive by ELISA. The tests were conducted at different times and concordance was not reported for paired blood samples. Da Silva Ferreira *et al.* reported 4/20 ML Flow negative and 3/20 NDO-LID negative (20/20 were PGL-1 ELISA and RLEP PCR positive), and suggested that these false negative lateral flow test results could be related to stage of infection because the false-negative animals showed lower anti-PGL-1 reactivity [22]. Da Silva *et al* used polyclonal rabbit antibody to localise PGL-1 antigen in spleen sections but did not report overall positivity in all the armadillos in their sample [3].

## Clinical signs

Four studies investigated clinical signs of leprosy, three with positive findings (Deps [23], Antunes [7], da Silva Ferreira *et al* [22]), one negative (de Souza [28]). Deps found head or body ulcers in 12% (6/52) and ulcerated lesions on the paws and/or internal carapace of 96% (50/52) of armadillos in a study with 53% (19/36) positivity by PCR [23]; none of the animals had a typical clinical picture of disseminated disease similar to human Virchowian leprosy [31]. In 65 armadillos in a study with 6% (4/65) PCR-positivity, Antunes identified clinical alterations in 95% (62), including external lesions (20%), lymphadenomegaly (49%), liver (30%) and splenic (35%) lesions, splenomegaly (27%) and hepatomegaly (24%) [7]. In a study which found 100% positivity by PCR in 20 armadillos, skin lesions were identified in 6 animals (30%), splenomegaly in 4 (20%) and lymphadenopathy in 7 (35%) [22]. Complete absence of clinical signs in 50 armadillos as reported by de Souza were consistent with the entirely negative PCR results in this study [28].

## Discussion

This is the first systematic review of natural *Mycobacterium leprae* infection in wild armadillos in Brazil, 17 years after the first report of *M. leprae* in nine-banded armadillos (*Dasypus*

*novemcinctus*) caught in the south-eastern state of Espírito Santo [6]. Our review shows that the prevalence of *M. leprae* in samples from armadillo populations in Brazil varies from 0% to 100% (pooled average 9.4%), that *M. leprae* infects the two main species (*Dasypus novemcinctus* and *Euphractus sexcinctus*), and that natural infection has been reported from the north and north eastern states of Pará [3], Ceará [8] and Rio Grande do Norte [22] and as far south as Espírito Santo state [7, 23, 26, 27]. Whether the negative findings of two studies conducted further to the south and east of Espírito Santo indicate a limit to the spread of *M. leprae* infection in armadillos is uncertain, although both studies were relatively large [9, 28]. Similarly, absence of *M. leprae* infection (except for a possible paucibacillary case determined by histopathology) in armadillos captured in Amazonas state, a Hansen's Disease endemic region, does not provide conclusive evidence of absence given the small sample size [10]. Indeed, relatively small sample sizes in all the included studies meant that the observed overall variation in *M. leprae* prevalence could be entirely attributable to sampling error [32], rather than to real variation i.e. some armadillo populations being heavily infected with *M. leprae* whilst others are disease-free or to artefactual variation, i.e. arising from differences in methods.

The most important methodological differences that could explain some of the observed variation in *M. leprae* prevalence as detected by RLEP PCR relate to biological samples (methods of specimen collection, processing and storage) and the presence of PCR inhibitors. Regarding the latter, the authors' own experience of using PCR to detect *M. leprae* DNA is that the amount of inhibitors varies considerably depending on sample methods, leading to false negative results. The use of positive controls (purified *M. leprae* DNA) does not solve the problem because it only gives certainty that the PCR reaction worked but does not detect inhibitors present in the sample. Instead it is necessary to make a control of inhibitors directly in the samples by reconstituting negative ones with *M. leprae* DNA and repeating the PCR. Inhibitory substances were tested for in only 3 of the 8 PCR studies [8, 10, 22], and although they were detected in only a small proportion of samples we cannot discount false negative results possible affecting prevalence estimates in the other 5 studies, two of which reported 0% *M. leprae* prevalence [9, 28].

With regard to single *vs.* nested PCR, the former method was used by the three studies which reported 0% *M. leprae* positivity [9, 10, 28] and by the study with the lowest non-zero prevalence [7]. However, all four studies used positive controls and da Silva Ferreira *et al*, in their study which found 20/20 positive, reported that nested PCR had detected only one additional positive animal after the first round of PCR [22]. All but one of the 8 PCR studies targeted RLEP, albeit with some differences in the primers used, therefore differences in *M. leprae* DNA targets are unlikely to account for overall variation. Also, the one study with a different target (65kDa) reported 53% positive samples [23]. Whilst these methodological differences might contribute to uncertainty in quantifying accurately *M. leprae* prevalence in a given armadillo population, they are unlikely to account for the very wide variation that we found in our review. However, we concur with da Silva Ferreira *et al* who argued that it would be of benefit to future studies in this area to standardize methods [22], with a protocol based on best practice in specimen collection, handling and processing and standardized PCR methods in terms of primers and testing for inhibitory substances.

Another methodological aspect in which we find ourselves in agreement with da Silva Ferreira *et al* is that simpler and more rapid methods for *M. leprae* testing in wild armadillos, such as the ML Flow and NDO-LID tests, should be considered for future studies. Indeed, the ML Flow test was first evaluated as being potentially suitable by one of our authors in 2007 [26]. We would argue that, depending on the scientific question being asked, the lower sensitivities of such tests might be outweighed by their ease of use and non-lethality. As with PCR, these methods would need to be standardized to ensure comparability between studies. Conversely,

a qualitative appraisal of results from the studies in our review suggests that histopathological methods and clinical examination for signs of leprosy are less useful. The former have the disadvantages of requiring tissue samples, being difficult to perform in the field, and being less sensitive than PCR as shown by discordant results in studies where both were performed. The discriminatory utility of clinical signs, although characterized to some extent in laboratory animals [33], is unknown in wild armadillos and is probably susceptible to observer bias (dependent on expertise and experience) and to selection bias (if animals with advanced disease are more or less likely to be caught).

Variation in the prevalence of *M. leprae* infection in armadillo populations in Brazil as a real natural phenomenon merits further investigation and requires studies very different in design from those reviewed here. Indeed, we would argue that further studies based on small (N<100) samples from selected locations are not going to further our understanding. Instead, much larger and longer-term studies conducted in partnership with national or regional animal conservation and ecology groups are needed to map *M. leprae* infection in armadillos across Brazil. At the same time, data must be gathered in studies focused on subpopulations of armadillos in endemic areas of Brazil to characterize *M. leprae* transmission and persistence within groups of animals, for example, using trackers and repeated sampling over the armadillo lifespan, as has been done in the USA to gather data which were then used to model *M. leprae* spread within the armadillo population [34]. Such studies could also test for other non-tuberculosis mycobacteria, including *M. lepromatosis*, the other causal agent of Hansen's Disease [35].

Variability of the prevalence among armadillos could be related to different habitats. The observed zero prevalence among armadillos in the Central-West Region could be a consequence of seasonal floods in the Pantanal that. This environment favours larger populations of wild animals, including armadillos, and representative samples would require many more animals to estimate the true prevalence of *M. leprae* infection in such areas.

Of note is that the Nine-banded armadillo, *Dasypus novemcinctus*, has tended to be the focus of studies regarding *M. leprae* in armadillos [33], representing 69% of our pooled sample and providing 7 prevalence estimates compared with 4 for *Euphractus sexcinctus*. However, data from a case-control study of Hansen's Disease risk in relation to armadillo contact [1] show that *Euphractus sexcinctus* was eaten almost as frequently (by 63% (94/149) of respondents) as *Dasypus novemcinctus* (74% (110/149)), *Priodontes maximus* by 12% (18/149) and *Tolypeutes tricinctus* by 11% (17/149)), and the recent study by da Silva Ferreira *et al* reported 100% *M. leprae* prevalence in 20 *Euphractus sexcinctus* [22]. A wide variety of contact with armadillos through hunting, cooking and consumption of armadillo meat was described among residents of the State of Ceará, in north-eastern Brazil [36]. The complexities of human-armadillo interaction in relation to *M. leprae* include the suggestion that transmission can occur in the opposite direction [29] and evidence that infection in armadillos is part of a wider environmental pool of *M. leprae* [37].

## Strengths and limitations

The main strength of our review is that its scope was very focused, and we are confident that all relevant studies have been identified, including 5 theses or dissertations [7, 23–25, 28]. Three of these were included in our review because they provided data that had not been published [7, 23, 28]; two were rated as 'good' quality. We were also able to include a very recent study which had not been indexed when the databases were searched [22]. Another strength is that the methods used by the included studies to obtain the estimates of *M. leprae* prevalence that we used in our main meta-analysis were reasonably homogeneous, i.e. PCR targeting

RLEP (*M. leprae*-specific repetitive element). The main limitation is that the relatively small sizes of the included studies (all but one of the eight studies contributing to the main PCR meta-analysis had ≤50 armadillos) combined with the range of prevalences (including several studies with zero positive animals) yielded a very wide predictive interval. This means that we cannot ascertain how much variability in *M. leprae* prevalence might be attributable to differences in methods or how much it represents real variation in *M. leprae* prevalence across armadillo populations in Brazil.

## Conclusion

The true risk to human health of contact with *M. leprae*-infected armadillos has not been systematically reviewed, but evidence from Brazil and other countries indicates an association between contact with armadillos and increased risk of Hansen's Disease [1–5]. Whilst Hansen's Disease is officially recognized as zoonotic in the USA, with recommendations regarding contact with armadillos [38], no recommendations have been made by the Ministry of Health in Brazil or the National Programme for Control of Hansen's Disease. The fraction of Hansen's Disease in the human population that is attributable to contact with armadillos will depend on the magnitude of the risk, the type and frequency of contact and consumption and how common these practices are in the population, the role of other (human-to-human) transmission routes for *M. leprae*, and the immunological susceptibility of the individual. Our review has shown that one other possible factor, the prevalence of *M. leprae* in wild armadillos, cannot be predicted with any certainty based on data from studies conducted to date, although average prevalence is equivalent to 1 in 10 armadillos in Brazil being infected. The large-scale long-term studies that we suggest for future research could attempt to correlate *M. leprae* in wild armadillos with proximity to human habitats. In the meantime, the precautionary principle should prevail, with public health and educational efforts directed towards improving community knowledge and changing behaviour to protect human and armadillo populations.

## Supporting information

**S1 Checklist. PRISMA checklist.**
(DOCX)

**S1 Fig. Funnel plot with pseudo 95% confidence limits of *M. leprae* prevalence in wild armadillos in Brazil detected using PCR methods (corresponding to Fig 4).**
(TIF)

**S1 Table. Detailed description of included studies investigating *M. leprae* infection in wild armadillos in Brazil.**
(DOCX)

**S1 Data. Supplementary Appendices (search terms, references after de-duplication, quality assessment (QA) tool, QA scores, extracted data).**
(XLSX)

**S1 Flow Diagram. PRISMA flow diagram.**
(TIF)

## Acknowledgments

The authors thank Professor Cecília Calabuig (Universidade Federal Rural do Semi-Árido, Brazil) for creating the map showing the study sites.

## Author Contributions

**Conceptualization:** Patrícia Deps, Simon M. Collin.

**Data curation:** Patrícia Deps, João Marcelo Antunes, Simon M. Collin.

**Formal analysis:** Simon M. Collin.

**Investigation:** Patrícia Deps, João Marcelo Antunes, Simon M. Collin.

**Methodology:** Patrícia Deps, João Marcelo Antunes, Adalberto Rezende Santos, Simon M. Collin.

**Validation:** Simon M. Collin.

**Visualization:** Simon M. Collin.

**Writing – original draft:** Patrícia Deps, Simon M. Collin.

**Writing – review & editing:** Patrícia Deps, João Marcelo Antunes, Adalberto Rezende Santos, Simon M. Collin.

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
