## [Decision Letter · Decision Letter 0]

6 Feb 2020

Dear Professor Deps,

Thank you very much for submitting your manuscript "Prevalence of Mycobacterium leprae in armadillos in Brazil: a systematic review and meta-analysis" for consideration at PLOS Neglected Tropical Diseases. As with all papers reviewed by the journal, your manuscript was reviewed by members of the editorial board and by several independent reviewers. The reviewers appreciated the attention to an important topic. Based on the reviews, we are likely to accept this manuscript for publication, providing that you modify the manuscript according to the review recommendations. 

Sincerely,

Carlos Franco-Paredes

Associate Editor

Gerd Pluschke

Deputy Editor

Reviewer's Responses to Questions

**Key Review Criteria Required for Acceptance?**

**Methods**

-Are the objectives of the study clearly articulated with a clear testable hypothesis stated?

-Is the study design appropriate to address the stated objectives?

-Is the population clearly described and appropriate for the hypothesis being tested?

-Is the sample size sufficient to ensure adequate power to address the hypothesis being tested?

-Were correct statistical analysis used to support conclusions?

-Are there concerns about ethical or regulatory requirements being met?

Reviewer #1: Yes, objective is clear and robust, as well as the method used. No ethical concerns to comment.

Reviewer #2: The methods are clearly articulated, the design is appropriate to address the objectives the authors proposed, the chosen criteria for the papers was clear, the sample size is a concern as acknowledged, the statistical analysis was appropriate and supports the conclusions, no concern related to ethical issues.

Reviewer #3: Prevalence of Mycobacterium leprae in armadillos in Brazil: a systematic review and meta-analysis

1 investigate natural M leprae infection in wild armadillos in endemic country to disease in human that have a control program but the new cases are maintenance and as armadillos are eaten in theses areas they can be infection to human and human to armadillos but quantify and explore the variability in prevalence of armadillos infection will be very difficult with this systemic review.

2. It is a systematic review protocol, but there are many limitations by the variability of methodology between the studies.

3. the armadillos live in endemic areas of hansen disease human and are animals susceptible to infection of M leprae, but as they no are domestic animals perhaps all sample had few size.

4 and 5. my knowledge is poor in meta-analysys, but the discussion and conclusions showed are supported by the revision.

6. the authors suggest new study only with blood and in few studies the armadillos were euthanised and probable when ethical or regulatory requirements were more light to experimental in animals.

**Results**

-Does the analysis presented match the analysis plan?

-Are the results clearly and completely presented?

-Are the figures (Tables, Images) of sufficient quality for clarity?

Reviewer #1: This is a systematic review and meta-analysis proposal. Results are clearly stated. Tables are necessary and help readers to clearly understand the method and findings.

Reviewer #2: The analysis matches the plan, the results are clearly presented, tables have clarity but images are of low quality, suggest replacement.

Reviewer #3: The analysis presented match the analysis plan with one systematic review protocol and the bibliographies included non-indexed citations as theses were justified

 line 113-4: M leprae was detected is important defined what is this? Same line 127: prevalence of M leprae.

Table 1 is better figure 1: 

 Deps 69 or 66 cases included 52 need more explain about this. line 151-3

 Antunes line 202 (2 or 4 /65)

 Frota 27 + 2 = 2

**Conclusions**

-Are the conclusions supported by the data presented?

-Are the limitations of analysis clearly described?

-Do the authors discuss how these data can be helpful to advance our understanding of the topic under study?

-Is public health relevance addressed?

Reviewer #1: To some extent, yes. The conclusion clearly state the main finding in what relates the proposed objective.

However, some digresion to the main objective seems to be unnecessary and somewhat tendentious to a subject leprosy as a zoonotic disease) not strictly relate to the main objective (see below).

Reviewer #2: The conclusion and limitations are well stated and discussed, as well as the meaning of their findings. The public health relevance of the study is also addressed.

Reviewer #3: Objectives of quantify and explore the variability in prevalence of armadillos infection will be very difficult with this systemic review, because of the limitations by the differences in methodology of the studies and because the definition of infection in leprosy is yet very difficult if the same methodology is used, excepted when the bacillos are found in tissue for stain specific but few studies used this methodology.

The suggestion of recognized the armadillos infection by Brazil govern is the most important conclusion of this research and the changing behaviour to protect human and armadillos.

**Editorial and Data Presentation Modifications?**

Reviewer #1: L. 72 - …sequelae of Hansen’s Disease are entirely avoidable if diagnosed and treated early [13]…

I would suggest to remove “entirely” – sometimes this is not true.

L. 74-77 - Whilst the proportion of new cases in Brazil attributable to zoonotic transmission is unknown, the persistence of armadillo capture and consumption as a cultural habit in some parts of Brazil means that the role of wild armadillos as a reservoir of M. leprae is relevant to efforts to eliminate the disease [16]. 

This paragraph has an intrinsic disruption. There is no sense to link “the persistence of armadillo capture and consumption” to the possibility of armadillos being or not a reservoir of M. lerprae and, chiefly, to link this to efforts to eliminate leprosy! This paragraph is equivocal and should be deleted.

Reviewer #2: I recommend minor revisions.

Reviewer #3: line 113-4: M leprae was detected is important defined better and line 127: prevalence of M leprae defined better.

Table 1 suggest figure 1: 

 Deps 69 or 66 cases included 52 need more explain better this. line 151-3

 Antunes line 202 (2 or 4 /65)

 Frota 27 + 2 = 2

the bacillos are found in tissue for stain specific but few studies used this methodology explain more about this limitations, or the definition of infection.

**Summary and General Comments**

Reviewer #1: The method used has enough strength to congratulate authors for their paper. The final paragraph in ”Strengths and Limitations” is relevant to the entire study. 

Indeed, difficult to ascertain how much the variability in prevalence might be attributed to differences in methods or how much it represents a real variation in the prevalence of M. leprae in the armadillos. 

This reported limitation recommends some extra caution in their conclusions: leprosy as a zoonosis in USA is a somewhat "political" declaration and to me it is a wise decision of the Brazilian health authorities not to take this seriously for Brazil. It seems that authors do not agree with this cautious and correct decision - however, the results of their review paper is in itself a strong reason for not doing so! 

Authors should be congratulated for intensively recommend new and lengthy studies in the attempt to correlate M. leprae in wild armadillos with proximity to human habitats. Meanwhile, considering their results & conclusions, let us not blame armadillos for playing any key role in the epidemiology of leprosy in Brazil – there are no clear evidences for that so far. In this sense, we would recommend authors to review some parts of their text (namely in Introduction and Conclusions) where a tendency to speak in favor of this thesis is strongly biased and intentionally suggested. Their main objective is to discuss prevalence of M. leprae in wild armadillos, not to claim that armadillos are responsible for the endemy of leprosy in Brazil, as they tend to suggest.

Reviewer #2: The authors properly discuss the variability in the prevalence of leprosy among the armadillos of the articles chosen. They propose laboratory tests that are appropriate and might be chosen and standardized in future studies.

The authors could discuss in some more considerable depth both the implications of their small sample size, and the great variability of prevalence among the armadillos from the different studies.

The authors fail to cite a study that mentions the possibilities of transmission between armadillos and humans. They also fail to mention/cite that the transmission may occur in the opposite direction as well.

In general, the article is very well written.

Reviewer #3: Hansen disease is very important public health in endemic countries as Brazil, that is in the first tax of detection new cases of the world and about 40 years ago have a specific and free multi-drug-therapy. As armadillos could be important to the infection to human and human to armadillos this revision can contribution to call attention of the govern and the researchers for the importance of studies in this theme neglected disease if treated early no disability and no transmission..

PLOS authors have the option to publish the peer review history of their article (what does this mean?). If published, this will include your full peer review and any attached files.

Reviewer #1: No

Reviewer #2: No

Reviewer #3: No
---

## [Editor Report · Decision Letter 1]

10 Feb 2020

Dear Professor Deps,

We are pleased to inform you that your manuscript 'Prevalence of Mycobacterium leprae in armadillos in Brazil: a systematic review and meta-analysis' has been provisionally accepted for publication in PLOS Neglected Tropical Diseases.

Before your manuscript can be formally accepted you will need to complete some formatting changes, which you will receive in a follow up email. A member of our team will be in touch within two working days with a set of requests.

Best regards,

Carlos Franco-Paredes

Associate Editor

Gerd Pluschke

Deputy Editor

---

## [Editor Report · Acceptance letter]

11 Mar 2020

Dear Professor Deps,

We are delighted to inform you that your manuscript, "Prevalence of *Mycobacterium leprae* in armadillos in Brazil: a systematic review and meta-analysis," has been formally accepted for publication in PLOS Neglected Tropical Diseases.

Best regards,

Serap Aksoy

Editor-in-Chief

Shaden Kamhawi

Editor-in-Chief
